# Automatic Induction of Synsets from a Graph of Synonyms

## Abstract

This paper presents a new graph-based approach that induces synsets using synonymy dictionaries and word embeddings. Firstly, we build a weighted graph of synonyms extracted from commonly available resources, such as Wiktionary. Secondly, we apply word sense induction to deal with ambiguous words. Finally, we cluster the disambiguated version of the ambiguous input graph into synsets. Our meta-clustering approach lets us use an efficient hard clustering algorithm to perform fuzzy clustering of the graph. Despite its simplicity, our approach shows excellent results outperforming five analogous state-of-the-art methods in terms of F-score on four different gold standard datasets for English and Russian derived from large-scale manually constructed lexical resources.

## 1 Introduction

A *synset* is a set of mutual synonyms, which can be represented as a clique where nodes are words and edges are synonymy relations. Synsets represent word senses and are building blocks of WordNet (Miller, 1995) and similar resources as thesauri and lexical ontologies. These resources are crucial for many natural language processing applications which require common sense reasoning, such as information retrieval (Gong et al., 2005) and question answering (Kwok et al., 2001; Zhou et al., 2013). However, for most languages no manually-constructed resource comparable to the English WordNet in terms of coverage and quality is available. For instance, Kiselev et al. (2015) presents a comparative analysis of lexical resources available for the Russian language. This

lack of linguistic resources for many languages urges the development of new methods for automatic construction of WordNet-like resources.

Wikipedia[1], Wiktionary[2], OmegaWiki[3] and other collaboratively-created resources contain a large amount of lexical semantic information—yet designed to be human-readable and not formally structured. While semantic relations can be automatically extracted using such tools as DKPro JWKTL[4] and Wikokit[5], words in these relations are not disambiguated. For instance, the synonymy pairs (*bank*, *streambank*) and (*bank*, *banking company*) will be connected via the word "bank", while they refer to the different senses. This problem stems from the fact that articles in Wiktionary and similar resources list undisambiguated synonyms. They are easy to disambiguate for humans while reading a dictionary article, but can be a source of errors for a language processing system.

The contribution of this paper is a novel approach which resolves ambiguities of the input graph enabling fuzzy clustering. The method takes as an input synonymy relations between potentially ambiguous terms available in human-readable dictionaries and transforms them into a disambiguated machine readable representation in the form of synsets. Our method, called WATSET, is based on a new meta-algorithm for fuzzy graph clustering. The name choice reflects the underlying principle: discover the correct word senses ("wat") and then construct the synsets ("set").

In contrast to the projects like BabelNet (Navigli and Ponzetto, 2012) and UBY (Gurevych et al., 2012), which rely on English WordNet as

---

[1] http://www.wikipedia.org
[2] http://www.wiktionary.org
[3] http://www.omegawiki.org
[4] https://dkpro.github.io/dkpro-jwktl
[5] https://github.com/componavt/wikokit

a pivot for mapping of existing resources, our approach requires no such pivot lexical ontologies. Besides, it outperforms analogous state-of-the-art methods for synset induction. An implementation of the WATSET method is available online, along with induced lexical resources.[6]

## 2 Related Work

**Methods based on resource mapping**, such as BabelNet and UBY, gather various existing lexical resources across multiple languages and perform their linking to obtain a machine-readable repository of lexical semantic knowledge. In its core, BabelNet was obtained by mapping the Princeton WordNet and Wikipedia enhanced by the machine translation of the results. Later, other resources were mapped to this core, including Wiktionary and OmegaWiki. UBY has a similar architecture, but relies on similarity of dictionary definitions and existing cross-lingual links for mapping.

The potential advantages of our approach as compared to BabelNet/UBY is the (1) absence of the error-prone procedures of mapping and machine translation; (2) possibility to model the target language more accurately as senses of the same word in different languages may be different; (3) no need for a pivot English resource.

A related branch of methods deals with coarsification of sense inventories of fine-grained lexical resources, such as (Snow et al., 2007).

**Methods based on word sense induction** try to induce sense representations without the need for any initial lexical resource by extracting semantic relations from text. In particular, word sense induction (WSI) based on word ego networks clusters graphs of words semantically related to the ambiguous word (Lin, 1998; Pantel and Lin, 2002; Dorow and Widdows, 2003; Véronis, 2004; Hope and Keller, 2013). Each cluster corresponds to a word sense. An ego network consists of a single node (ego) together with the nodes they are connected to (alters) and all the edges among those alters. In the case of WSI, such a network is a local neighbourhood of one word. Nodes of the ego network are the words which are semantically similar to the target word.

Such approaches are able to discover homonymous senses of words, e.g., "bank" as slope versus "bank" as organisation (Di Marco and Navigli, 2012). However, as the graphs are composed of semantically related words obtained using distributional methods (Baroni and Lenci, 2010; Biemann and Riedl, 2013), the resulting clusters by no means can be considered synsets. Namely, (1) they contain words related not only via synonymy relation, but via a mixture of relations such as synonymy, hypernymy, co-hyponymy, antonymy, etc. (Heylen et al., 2008); (2) clusters are not unique, i.e., one word can occur in several clusters referring to the same sense, while in WordNet a word used in the given sense occurs only in a single synset.

In our approach, to induce synsets, we use word ego network clustering similarly as in word sense induction approaches, but apply them to the graph of semantically clean synonyms.

**Methods based on clustering of synonyms**, such as our approach, induce the resource from an ambiguous graph of synonyms where edges were extracted from manually-created resources. According to the best of our knowledge, most experiments either used graph-based word sense induction applied to text-derived graphs or used a mapping-based method which already assumes availability of a WordNet-like resource. A notable exception is the ECO approach by Gonçalo Oliveira and Gomes (2014), which was used to induce a WordNet of the Portuguese language called Onto.PT.[7] We use this approach and five other state-of-the-art graph clustering algorithms as the baselines:

- **ECO** (Gonçalo Oliveira and Gomes, 2014) is a fuzzy clustering algorithm that was used to induce synsets of a Portuguese WordNet out of available synonymy dictionaries. The algorithm starts by adding random noise to edge weights. Then, the approach launches Markov Clustering of this graph several times to estimate the probability of each word pair being in the same synset. Finally, these candidate pairs passing through the $\theta$ threshold are added to output synsets.

- **MaxMax** (Hope and Keller, 2013) is a fuzzy clustering algorithm designed initially for the word sense induction task. In the nutshell, pairs of nodes are grouped if either node has a maximal affinity to the other. The algorithm starts by converting the input undirected graph into the directed graph by keep-

---

[6]scheme://domain.tld/anonymized

[7]http://ontopt.dei.uc.pt

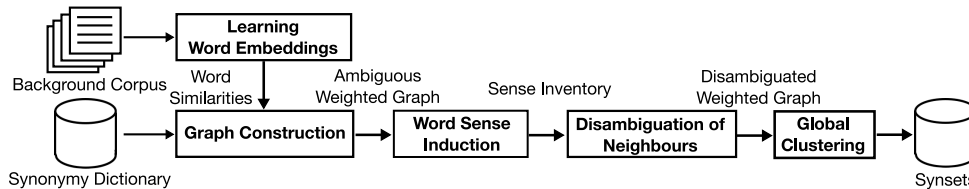

Figure 1: Outline of the WATSET method for synset induction.

ing maximal nodes of each node. Next, all nodes are marked as root nodes. Finally, for each root node, the following procedure is repeated: all transitive children of this root form a cluster and marked as non-root nodes; a root node together with all its transitive children form a fuzzy cluster.

- **Markov Clustering** (MCL) (van Dongen, 2000) is a hard clustering algorithm for graphs based on simulation of stochastic flow in graphs. MCL simulates random walks within a graph by alternation of two operators called expansion and inflation, which recomputes the class labels. Notably, it has been successfully used for the word sense induction task (Dorow and Widdows, 2003).

- **Chinese Whispers** (Biemann, 2006) is a hard clustering algorithm for weighted graphs that can be considered as a special case of MCL with a simplified class update step. At each iteration, the labels of all the nodes are updated according to the majority labels among the neighbouring nodes. The algorithm has a meta-parameter that controls graph weights that can be set to three values: (1) *top* sums over the neighbourhood's classes; (2) *nolog* downgrades the influence of a neighbouring node by its degree or (3) *log* of its degree.

- **Clique Percolation Method** (CPM) (Palla et al., 2005) is a fuzzy clustering algorithm for unweighted graphs that builds up the clusters from $k$-cliques corresponding to fully connected sub-graphs of $k$ nodes. While this method is only commonly used in social network analysis, we decided to add it to the comparison as synsets are essentially cliques of synonyms, and it is natural to try using an algorithm based on detection of cliques.

## 3   The WATSET Method

The goal of the method is to induce a set of unambiguous synsets by grouping individual ambiguous synonyms. An outline of the proposed approach is depicted in Figure 1. The method takes a dictionary of ambiguous synonymy relations and a text corpus as an input and outputs synsets. Note that the method can be used without a background corpus, yet as our experiments will show, corpus-based information improves the results, when utilizing it for weighting the word graph.

A synonymy dictionary can be perceived as a graph, where the nodes correspond to lexical entries (words) and the edges connect pairs of the nodes when the synonymy relation between them holds. The cliques in such a graph naturally form densely connected sets of synonyms corresponding to concepts (Kamps et al., 2004). Given the fact that the clique problem in a graph is NP-complete (Bomze et al., 1999), an efficient graph clustering algorithm like the MCL algorithm can be used for finding a global segmentation of the graph. However, the hard clustering property of this algorithm does not handle polysemy: one word can have several senses but will be assigned to only one cluster. To deal with this limitation, a word sense induction procedure is used to induce senses for all words. Finally, the disambiguated word sense graph is clustered globally to induce the synsets from this disambiguated word graph.

More specifically, the method consists of five steps presented in Figure 1: (1) learning word embeddings; (2) constructing the ambiguous weighted graph of synonyms $G$; (3) inducing the word senses; (4) constructing the disambiguated weighted graph $G'$ by disambiguating of neighbours w.r.t. the induced word senses; (5) global clustering of the graph $G'$.

### 3.1   Learning Word Embeddings

Since the different graph clustering algorithms are sensitive to edge weighing, we consider the distributional semantic similarity measures based on word embeddings as a possible edge weighing approach for our synonymy graph. As we show further, this approach yields the best results.

### 3.2 Construction of a Synonymy Graph

We construct the synonymy graph $G = (V, E)$ as follows. The set of nodes $V$ includes every lexeme appearing in the input synonymy dictionaries. The set of undirected edges $E$ is composed of all edges $(u, v) \in V \times V$ retrieved from one of the input synonymy dictionaries. We consider three edge weight representations:

- **ones** that assigns every edge the constant weight of 1;

- **count** that weights the edge $(u, v)$ as the number of times the synonymy pair appeared in the input dictionaries;

- **sim** that assigns every edge $(u, v)$ a weight equal to the cosine similarity of the skip-gram word vectors.

As the graph $G$ is likely to have polysemous words, the goal is to separate the individual word senses using graph-based word sense induction.

### 3.3 Word Sense Induction

We use a graph-based word sense induction method that is similar to the curvature-based approach of Dorow and Widdows (2003). In particular, removal of the nodes participating in many triangles tends to separate the original graph into several connected components. Thus, given a word $u$, we extract a network of its nearest neighbours from the synonymy graph $G$. Then, we remove the original word $u$ from this network and run a hard graph clustering algorithm that assigns one node to one and only one cluster. In our experiments, we test Chinese Whispers and Markov Clustering. As the result, each cluster is hopefully representing a different sense of the word $u$, e.g.:

$bank^1$ {*streambank*, *riverbank*, ... }
$bank^2$ {*bank company*, ... }
$bank^3$ {*bank building*, *building*, ... }
$bank^4$ {*coin bank*, *penny bank*, ... }

We denote $man^1$, $man^2$ and other items as word senses referred to as, e.g., $\mathrm{senses}(\mathrm{man})$. We denote as $\mathrm{ctx}(s)$ a cluster corresponding to the word sense $s$. Note that the context words have no sense labels. They are recovered by the disambiguation approach described below.

### 3.4 Disambiguation of Neighbours

The result of the previous step is splitting word nodes into (one or more) sense nodes. However, nearest neighbours of each new node are still ambiguous, e.g., $(bank^3, building^?)$.

For recovering these sense labels of the neighbouring words, we employ the following sense disambiguation approach. For each word $u$ in the context $\mathrm{ctx}(s)$ of the sense $s$, we estimate the most similar sense of that word $\hat{u}$ using the cosine similarity measure between the context of the sense $s$ and the context of the candidate sense $u'$ in a vector space model:

$$\hat{u} = \arg\max_{u' \in \mathrm{senses}(u)} \cos(\mathrm{ctx}(s), \mathrm{ctx}(u')).$$

This approach makes it possible to construct a disambiguated context $\widehat{\mathrm{ctx}}(s)$ that corresponds to the certain word senses appearing in $\mathrm{ctx}(s)$:

$$\widehat{\mathrm{ctx}}(s) = \{\hat{u} : u \in \mathrm{ctx}(s)\}.$$

### 3.5 Global Clustering

Finally, we construct the word sense graph $G' = (V', E')$ using the disambiguated senses instead of the regular words and establishing the edges between these disambiguated senses:

$$V' = \{s : u \in V \wedge s \in \mathrm{senses}(u)\},$$
$$E' = \{(s, \hat{u}) : s \in V' \wedge \hat{u} \in \widehat{\mathrm{ctx}}(s)\}.$$

As the result, running a hard clustering algorithm on $G'$ produces the desired set of synsets.

Figure 2 illustrates the process of disambiguation of an input ambiguous graph on the example of the word "bank". As one may observe, disambiguation of the nearest neighbours is a necessity to be able to construct a global version of the sense-aware graph. Note that current approaches to WSI, e.g., (Véronis, 2004; Biemann, 2006; Hope and Keller, 2013), do not perform this step, but perform only local analysis of the graph.

## 4 Evaluation

We conduct our experiments on the data for two different languages. We evaluate our approach on two datasets for English to demonstrate its performance on a resource-rich language. Additionally, we evaluate it on two Russian datasets since Russian is a good example of an under-resourced language with a clear need for synset induction.

### 4.1 Gold Standard Datasets

For each language, we used two differently constructed lexical semantic resources listed in Table 1 to obtain gold standard synsets.

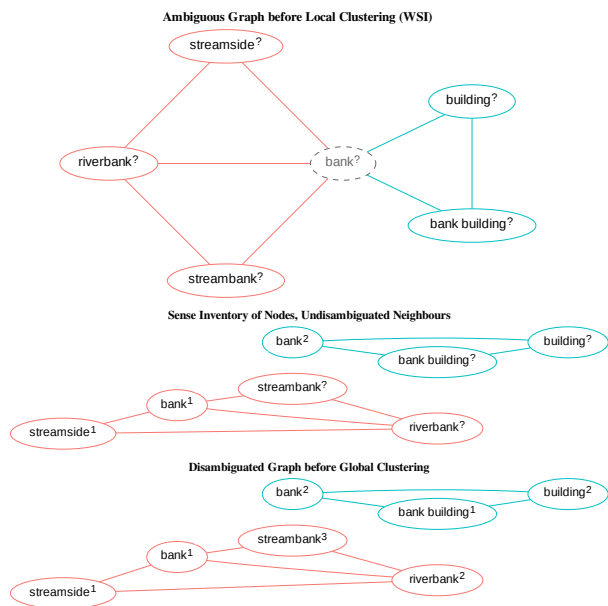

Figure 2: Disambiguation of an input ambiguous graph using local clustering (WSI) to facilitate global clustering of words into synsets.

**English.** We use **WordNet**[8] (Miller, 1995), a popular English lexical database crafted by a group of expert lexicographers. WordNet contains general vocabulary and appears to be *de facto* gold standard in similar tasks. We used WordNet 3.1 to derive the synonymy pairs from the synsets for the evaluation. Also, we use **BabelNet**[9] (Navigli and Ponzetto, 2012), a large-scale multilingual semantic network constructed automatically using WordNet, Wikipedia and other resources. We retrieved all the synonymy pairs from the Babel-Net 3.7 synsets that were marked as English. We also considered using TWSI by Biemann (2013), but found that it contains mostly hypernymy and co-hyponymy relations instead of being a viable source of synonymy information.

**Russian.** We use **RuThes**[10] (Loukachevitch, 2011), a lexical ontology for Russian containing both general vocabulary and domain-specific synsets related to sport, finance, economics, etc. Up to a half of the words in this resource are multi-word expressions (Kiselev et al., 2015), which is due to the coverage of domain-specific vocabulary. RuThes is constructed in the traditional way, namely by a small group of expert lexicog-

---

[8] https://wordnet.princeton.edu/
[9] http://babelnet.org/
[10] http://www.labinform.ru/pub/ruthes

raphers. Also, we use **Yet Another RussNet**[11] (**YARN**) by Braslavski et al. (2016) as yet another gold standard for Russian. The resource is constructed using crowdsourcing and mostly covers general vocabulary. Particularly, non-expert users are allowed to edit synsets in a collaborative way loosely supervised by a team of project curators. Due to the ongoing development of the resource, we selected as the gold standard only those synsets that were edited at least eight times in order to filter out noisy incomplete synsets.

| Resource | | # words | # synsets | # synonyms |
|---|---|---|---|---|
| WordNet | En | 148 730 | 117 659 | 152 254 |
| BabelNet | En | 11 710 137 | 6 667 855 | 28 822 400 |
| RuThes | Ru | 119 836 | 31 528 | 474 537 |
| YARN | Ru | 9 141 | 2 210 | 48 291 |

Table 1: Statistics of the gold standard datasets.

## 4.2 Evaluation Metrics

To evaluate the quality of the induced synsets, we transformed them into binary synonymy relations and computed precision, recall, and F-score on the basis of the overlap of these binary relations with the binary relations from the gold standard datasets. Given a synset uniting $n$ words, we generate a set of $\frac{n(n-1)}{2}$ pairs of synonyms.

## 4.3 Word Embeddings

**English.** We use the standard 300-dimensional word embeddings trained on the 100 billion tokens Google News corpus (Mikolov et al., 2013).

**Russian.** We use the 500-dimensional word embeddings trained using the skip-gram model with negative sampling (Mikolov et al., 2013) using a context window size of 10 with the minimal word frequency of 5 on a 12.9 billion tokens corpus of books lib.rus.ec. These embeddings were shown to produce state-of-the-art results for Russian in the RUSSE shared task[12] and are part of the Russian Distributional Thesaurus[13] (RDT).

## 4.4 Input Dictionary of Synonyms

For each language, we constructed a synonymy graph using openly available language resources. The statistics of the graphs used as the input in the further experiments are shown in Table 2.

---

[11] https://russianword.net/en/
[12] http://www.dialog-21.ru/en/evaluation/2015/semantic_similarity
[13] http://russe.nlpub.ru/downloads

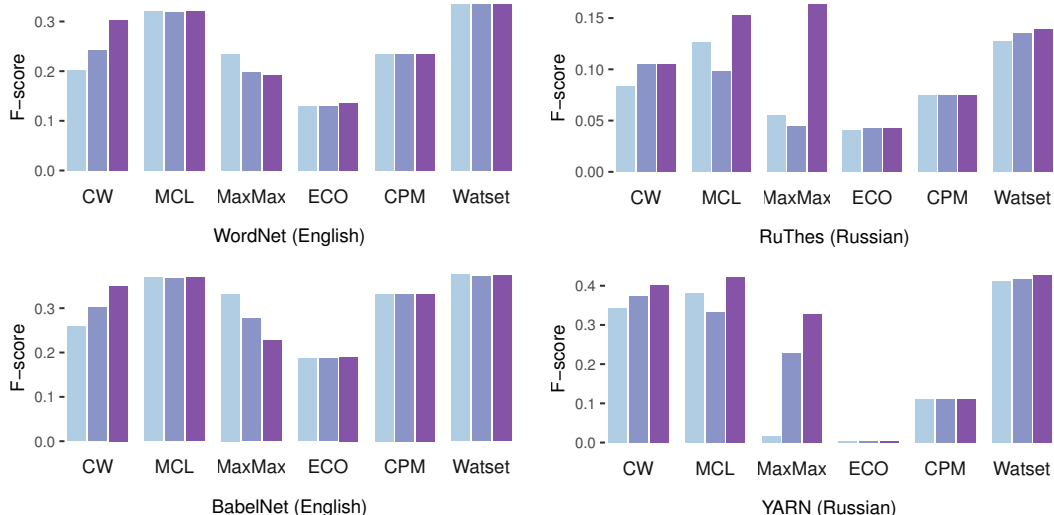

Figure 3: Impact of the different graph weighting schemas on the performance of synset induction: ■ ones, ■ count, ■ sim. Here, each bar corresponds to the top performance of each method.

**English.** The synonyms were extracted from the English Wiktionary, which is the largest Wiktionary as the present moment according to the lexical coverage, using the DKPro JWKTL tool by Zesch et al. (2008). Only the words marked as English have been extracted.

**Russian.** The synonyms from three sources were combined to improve lexical coverage of the input dictionary and to enforce confidence in jointly observed synonyms: (1) synonyms listed in the Russian Wiktionary extracted using the Wikokit tool by Krizhanovsky and Smirnov (2013); (2) the dictionary of Abramov (2007); and (3) the Universal Dictionary of Concepts (Dikonov, 2013). While these two latter resources are specific to Russian, Wiktionary is available for most languages. Note that the same input synonymy dictionary was used by authors of YARN to construct synsets using crowdsourcing. Therefore, results on the YARN dataset show how close an automatic synset induction method can approximate manually created synsets provided the same starting raw linguistic materials.

| Language | # words | # synonyms |
|----------|---------|------------|
| English | 77 871 | 71 816 |
| Russian | 74 395 | 202 313 |

Table 2: Statistics of the input datasets.

## 5 Results

We compare WATSET with five state-of-the art graph clustering methods presented in Section 2: Chinese Whispers (CW), Markov Clustering (MCL), MaxMax, ECO clustering, and the clique percolation method (CPM). In our experiments, we rely on our own implementation of MaxMax and ECO as reference implementations are not available. For CW[14], MCL[15] and CPM[16], the available implementations have been used. During the evaluation, we delete the clusters equal or larger than the threshold of 150 words as they hardly can represent any meaningful synset. The notation WATSET[MCL, CW$_{top}$] means using MCL for word sense induction and Chinese Whispers in the *top* mode for global clustering.

### 5.1 Impact of Graph Weighting Schema

Figure 3 presents an overview of the evaluation results on both datasets. The first step being common for all of the tested synset induction methods is the graph construction. Thus, we started with an analysis of the three ways to weight edges of the graph introduced in Section 3.2: binary scores (*ones*), frequencies (*count*), and semantic similarity scores (*sim*). The results across various configurations and methods indicate that using the weights based on the similarity scores provided by word embeddings is the best strategy for all the methods but MaxMax on both English datasets. However, in these cases, its performance using the *ones* weighing does not exceed the other methods using the *sim* weighing. Therefore, we report all further results on the basis of the *sim* weights.

---

[14] https://github.com/tudarmstadt-lt/chinese-whispers
[15] http://java-ml.sourceforge.net
[16] https://networkx.github.io

| Method | # words | # synsets | # synonyms | WordNet | | | BabelNet | | |
|---|---|---|---|---|---|---|---|---|---|
| | | | | Precision | Recall | F-score | Precision | Recall | F-score |
| WATSET[CW$_{top}$, MCL] | 77 871 | 55 874 | 78 303 | 0.290 | **0.397** | **0.335** | 0.272 | **0.478** | 0.346 |
| WATSET[CW$_{top}$, CW$_{top}$] | 77 871 | 55 873 | 78 337 | 0.290 | 0.396 | **0.335** | 0.271 | **0.478** | 0.346 |
| WATSET[MCL, MCL] | 77 871 | 33 948 | 127 331 | 0.328 | 0.334 | **0.331** | 0.340 | 0.414 | **0.374** |
| WATSET[MCL, CW$_{log}$] | 77 871 | 32 843 | 143 120 | **0.341** | 0.305 | 0.322 | 0.350 | 0.379 | **0.364** |
| MCL | 77 871 | 25 467 | 141 042 | 0.316 | 0.327 | 0.321 | 0.343 | 0.399 | **0.369** |
| CW$_{nolog}$ | 77 871 | 24 220 | 170 499 | **0.338** | 0.276 | 0.304 | **0.357** | 0.343 | 0.350 |
| CPM$_{k=2}$ | 54 374 | 19 749 | 80 325 | 0.148 | **0.551** | 0.234 | 0.243 | **0.525** | 0.332 |
| MaxMax | 77 871 | 25 532 | 436 571 | **0.363** | 0.131 | 0.192 | **0.380** | 0.163 | 0.228 |
| ECO | 77 870 | 56 152 | 25 726 | 0.075 | **0.765** | 0.136 | 0.109 | **0.717** | 0.190 |

Table 3: Comparison of the synset induction methods on datasets for English. All methods rely on the similarity edge weighting (*sim*); best configurations of each method in terms of F-scores are shown for each dataset. Results are sorted by F-score on WordNet, top three values of each metric are boldfaced.

| Method | # words | # synsets | # synonyms | RuThes | | | YARN | | |
|---|---|---|---|---|---|---|---|---|---|
| | | | | Precision | Recall | F-score | Precision | Recall | F-score |
| WATSET[CW$_{log}$, MCL] | 74 395 | 48 484 | 317 040 | **0.224** | 0.101 | **0.139** | 0.466 | 0.396 | **0.428** |
| WATSET[MCL, MCL] | 74 395 | 32 407 | 378 769 | 0.220 | 0.093 | 0.131 | 0.459 | 0.400 | **0.427** |
| WATSET[CW$_{nolog}$, CW$_{nolog}$] | 74 395 | 48 443 | 327 125 | **0.225** | 0.097 | 0.136 | **0.471** | 0.386 | **0.424** |
| MCL | 74 395 | 19 277 | 328 789 | 0.187 | 0.128 | **0.152** | 0.345 | 0.546 | 0.423 |
| WATSET[MCL, CW$_{nolog}$] | 74 395 | 31 129 | 429 823 | **0.231** | 0.084 | 0.123 | **0.500** | 0.352 | 0.413 |
| CW$_{nolog}$ | 74 395 | 16 683 | 660 424 | 0.220 | 0.069 | 0.105 | 0.460 | 0.357 | 0.402 |
| MaxMax | 74 395 | 24 250 | 560 096 | 0.186 | **0.147** | **0.164** | 0.230 | **0.566** | 0.327 |
| CPM$_{k=3}$ | 14 662 | 3 729 | 46 007 | 0.047 | **0.189** | 0.075 | 0.061 | **0.561** | 0.110 |
| ECO | 74 395 | 61 126 | 15 784 | 0.022 | **0.630** | 0.042 | 0.002 | **0.898** | 0.004 |

Table 4: Results on Russian sorted by F-score on YARN, top three values of each metric are boldfaced.

## 5.2 Performance Analysis

Table 3 and 4 present the evaluation results on the all the datasets for both languages. For each method, we show the best configurations in terms of F-score. One may note that the granularity of the resulting synsets, especially on the Russian dataset, is very different, ranging from 3 729 synsets for the CPM$_{k=3}$ method to 61 126 induced by the ECO method. Both tables report the number of words, synsets and synonyms after pruning huge clusters larger than 150 words. Without this pruning, the MaxMax and CPM methods tend to discover giant components obtaining almost zero precision as we generate all possible pairs of nodes in such clusters. The other methods did not demonstrate such behavior.

WATSET robustly outperformed all other methods according to F-score on both English datasets (Table 3) and on the YARN dataset for Russian (Table 4). Also, it outperformed all other methods according to precision on both Russian datasets. The disambiguation of the input graph performed by the WATSET method splits nodes belonging to several local communities to several nodes significantly facilitating the clustering task otherwise complicated by the presence of the hubs that wrongly link semantically unrelated nodes.

Amusingly, in all the cases, the toughest competitor was a hard clustering algorithm—MCL (van Dongen, 2000). We observed that the "plain" MCL successfully groups the monosemous words, but produces nonsensical clusters of moderate size, e.g., 20–50 words, when a polysemous word appears, resulting in the precision drop w.r.t. WATSET. Chinese Whispers, a simplified version of MCL, converges faster due to node label randomization which leads to a stricter stopping condition. CW thus does not amplify the hubs between the unrelated nodes and therefore produces smaller clusters in average. As the result, the "plain" CW offers higher precision than the "plain" MCL at the cost of lower recall.

Using CW instead of MCL for word sense induction in WATSET expectedly produces fine-grained senses. Interestingly, at the global clustering step, these senses erroneously tend to form coarse-grained synsets connecting unrelated senses of the ambiguous words. This explains the generally higher precision of WATSET[MCL, ·].

The MaxMax algorithm showed mixed results. On the one hand, it outputs large clusters uniting more than hundred nodes. This inevitably leads to a high recall, as it is clearly seen in the results for Russian because such synsets passed through our threshold of 150 words. The synsets produced on English datasets were even larger and did not pass, which resulted in low recall. On the other hand, the smaller synsets having at most 10–15 words were identified correctly. MaxMax appeared to be extremely sensible to the edge weighing, which complicates its practical use.

The CPM algorithm showed unsatisfactory results which indicated by emitting giant compo-

nents uniting thousands words. Such clusters have been automatically pruned, but the rest clusters connect virtually every node left after the pruning. This is confirmed by the high values of recall. As one increase the minimal number of elements in the clique $k$, precision effectively grows, but at the cost of a dramatic drop in recall. We suppose that the network structure assumptions exploited by CPM do not accurately model the structure of the synonymy graphs in the present task.

Finally, the ECO method yielded the worst results because the most cluster candidates failed to pass through the constant threshold used for estimating whether a pair of words should be included in the same cluster. Most synsets produced by this method were trivial, i.e., containing only one word. The remaining synsets for both languages have at most three words having been connected by a chance due to the edge noising procedure used in this method resulting in low precision.

## 6 Discussion

**On difference in absolute scores.** While the results obtained for the English datasets are in agreement (Tables 3 and 5), the results measured on RuThes and YARN datasets for Russian show a similar global picture in terms of relevant ranking of the methods, yet absolute scores are largely different (cf. Figure 3). This difference stems from the difference of the gold standards. RuThes is more domain-specific in terms of vocabulary, so our input set of generic synonymy dictionaries has a limited coverage on this dataset. On the other hand, recall calculated on YARN reaching significantly higher levels as this resource was manually built on the basis of exactly the same initial resources. Low performance of the cross-evaluation of two resources presented in Table 5 confirms this observation. No single resource for Russian can obtain high precision scores on another one. Surprisingly, even BabelNet, which integrates most of available lexical resources, still does not reach recall largely higher than $0.5$.[17] Finally, note that the results of this cross-dataset evaluation are not directly comparable to results in Table 4 as in our experiments we use much smaller input dictionaries than those used by BabelNet.

**On sparseness of the input dictionary.** Table 6 presents some examples of the obtained synsets

---

[17]We used BabelNet 3.7 as in our study for English and retrieved all 3 497 327 synsets that were marked as Russian.

| Resource | | Precision | Recall | F-score |
|---|---|---|---|---|
| BabelNet on WordNet | En | 0.998 | 0.729 | 0.843 |
| TWSI on WordNet | En | 0.036 | 0.021 | 0.027 |
| WordNet on BabelNet | En | 0.699 | 0.998 | 0.822 |
| TWSI on BabelNet | En | 0.046 | 0.027 | 0.034 |
| WordNet on TWSI | En | 0.015 | 0.539 | 0.030 |
| BabelNet on TWSI | En | 0.027 | 0.453 | 0.051 |
| YARN on RuThes | Ru | 0.110 | 0.142 | 0.124 |
| BabelNet on RuThes | Ru | 0.303 | 0.332 | 0.316 |
| RuThes on YARN | Ru | 0.118 | 0.512 | 0.192 |
| BabelNet on YARN | Ru | 0.105 | 0.515 | 0.174 |

Table 5: Performance of lexical resources cross-evaluated against each other.

of various sizes for the best WATSET configuration on both languages. As one might observe, the quality of the results is highly plausible. However, one limitation of all approaches considered in this paper is the dependence on completeness of the input dictionary of synonyms. In some parts of the input synonymy graph, important bridges between words can be missing, leading to smaller-than-desired synsets. A promising extension of the present methodology is using the distributional models to enhance connectivity of the graph to further improve recall of the method by adding extra relations.

| Size | Synset |
|---|---|
| 2 | {*decimal point, dot*} |
| 3 | {*gullet, throat, food pipe*} |
| 4 | {*microwave meal, ready meal, TV dinner, frozen dinner*} |
| 5 | {*objective case, accusative case, oblique case, object case, accusative*} |
| 6 | {*radio theater, dramatized audiobook, audio theater, radio play, radio drama, audio play*} |

Table 6: Sample synsets induced by the WATSET[MCL, MCL] method for English.

## 7 Conclusion

In this paper, we presented a new robust approach to fuzzy graph clustering that relies on a hard clustering method. Using ego network clustering, the nodes belonging to several local communities are split into several nodes each belonging to one local community. The transformed "disambiguated" graph is then clustered using an efficient hard graph clustering algorithm, obtaining a fuzzy clustering as the result. The disambiguated graph contains fewer hubs connecting unrelated nodes from different communities and thus facilitates clustering. We apply this meta-clustering algorithm to the task of synset induction, obtaining the best results on two datasets for two different natural language in terms of precision and competitive results in terms of F-score, as compared to five state-of-the-art graph clustering methods.

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
