# Peer review of "Automatic Induction of Synsets from a Graph of Synonyms"

_ACL 2017 — decision unknown_

[Official Review · Reviewer 1 · rating 4 · confidence 4]
soundness 3 · originality 4 · clarity 5 · impact 4 · substance 4 · appropriateness 5 · meaningful comparison 4 · presentation format Oral Presentation

This paper presents a graph-based approach for producing sense-disambiguated
synonym sets from a collection of undisambiguated synonym sets.  The authors
evaluate their approach by inducing these synonym sets from Wiktionary and from
a collection of Russian dictionaries, and then comparing pairwise synonymy
relations (using precision, recall, and F1) against WordNet and BabelNet (for
the English synonym sets) or RuThes and Yet Another RussNet (for the Russian
synonym sets).

The paper is very well written and structured.              The experiments and
evaluations
(or at least the prose parts) are very easy to follow.              The methodology
is
sensible and the analysis of the results cogent.  I was happy to observe that
the objections I had when reading the paper (such as the mismatch in vocabulary
between the synonym dictionaries and gold standards) ended up being resolved,
or at least addressed, in the final pages.

The one thing about the paper that concerns me is that the authors do not seem
to have properly understood the previous work, which undercuts the stated
motivation for this paper.

The first instance of this misunderstanding is in the paragraph beginning on
line 064, where OmegaWiki is lumped in with Wiktionary and Wikipedia in a
discussion of resources that are "not formally structured" and that contain
"undisambiguated synonyms".  In reality, OmegaWiki is distinguished from the
other two resources by using a formal structure (a relational database) based
on word senses rather than orthographic forms.              Translations, synonyms,
and
other semantic annotations in OmegaWiki are therefore unambiguous.

The second, and more serious, misunderstanding comes in the three paragraphs
beginning on lines 092, 108, and 120.  Here the paper claims that both BabelNet
and UBY "rely on English WordNet as a pivot for mapping of existing resources"
and criticizes this mapping as being "error-prone".  Though it is true that
BabelNet uses WordNet as a pivot, UBY does not.  UBY is basically a
general-purpose specification for the representation of lexical-semantic
resources and of links between them.  It exists independently of any given
lexical-semantic resource (including WordNet) and of any given alignment
between resources (including ones based on "similarity of dictionary
definitions" or "cross-lingual links").  Its maintainers have made available
various databases adhering to the UBY spec; these contain a variety of
lexical-semantic resources which have been aligned with a variety of different
methods.  A given UBY database can be *queried* for synsets, but UBY itself
does not *generate* those synsets.  Users are free to produce their own
databases by importing whatever lexical-semantic resources and alignments
thereof are best suited to their purposes.  The three criticisms of UBY on
lines 120 to 125 are therefore entirely misplaced.

In fact, I think at least one of the criticisms is not appropriate even with
respect to BabelNet.  The authors claim that Watset may be superior to BabelNet
because BabelNet's mapping and use of machine translation are error-prone.  The
implication here is that Watset's method is error-free, or at least
significantly less error-prone.  This is a very grandiose claim that I do not
believe is supported by what the authors ought to have known in advance about
their similarity-based sense linking algorithms and graph clustering
algorithms, let alone by the results of their study.  I think this criticism
ought to be moderated.              Also, I think the third criticism (BabelNet's
reliance
on WordNet as a pivot) somewhat misses the point -- surely the most important
issue to highlight isn't the fact that the pivot is English, but rather that
its synsets are already manually sense-annotated.

I think the last paragraph of §1 and the first two paragraphs of §2 should be
extensively revised. They should focus on the *general* problem of generating
synsets by sense-level alignment/translation of LSRs (see Gurevych et al., 2016
for a survey), rather than particularly on BabelNet (which uses certain
particular methods) and UBY (which doesn't use any particular methods, but can
aggregate the results of existing ones).  It may be helpful to point out
somewhere that although alignment/translation methods *can* be used to produce
synsets or to enrich existing ones, that's not always an explicit goal of the
process.  Sometimes it's just a serendipitous (if noisy) side-effect of
aligning/translating resources with differing granularities.

Finally, at several points in the paper (lines 153, 433), the "synsets" of TWSI
of JoBimText are criticized for including too many words that are hypernyms,
co-hypnomyms, etc. instead of synonyms.  But is this problem really unique to
TWSI and JoBimText?  That is, how often do hypernyms, co-hypernyms, etc. appear
in the output of Watset?  (We can get only a very vague idea of this from
comparing Tables 3 and 5, which analyze only synonym relations.)  If Watset
really is better at filtering out words with other semantic relations, then it
would be nice to see some quantitative evidence of this.

Some further relatively minor points that should nonetheless be fixed:

* Lines 047 to 049: The sentence about Kiselev et al. (2015) seems rather
useless.  Why bother mentioning their analysis if you're not going to tell us
what they found?

* Line 091: It took me a long time to figure out how "wat" has any relation to
"discover the correct word sense".  I suppose this is supposed to be a pun on
"what".  Maybe it would have been better to call the approach "Whatset"?  Or at
least consider rewording the sentence to better explain the pun.

* Figure 2 is practically illegible owing to the microscopic font.  Please
increase the text size!

* Similarly, Tables 3, 4, and 5 are too small to read comfortably.  Please use
a larger font.              To save space, consider abbreviating the headers ("P,
"R",
"F1") and maybe reporting scores in the range 0–100 instead of 0–1, which
will eliminate a leading 0 from each column.

* Lines 517–522: Wiktionary is a moving target.  To help others replicate or
compare against your work, please indicate the date of the Wiktionary database
dump you used.

* Throughout: The constant switching between Times and Computer Modern is
distracting.  The root of this problem is a longstanding design flaw in the ACL
2017 LaTeX style file, but it's exacerbated by the authors' decision to
occasionally set numbers in math mode, even in running text.  Please fix this
by removing

\usepackage{times}

from the preamble and replacing it with either

\usepackage{newtxtext}
\usepackage{newtxmath}

or

\usepackage{mathptmx}

References:

I Gurevych, J. Eckle-Kohler, and M. Matuschek, 2016. Linked Lexical Knowledge
Bases: Foundations and Applications, volume 34 of Synthesis Lectures on Human
Language Technologies, chapter 3: Linking Algorithms, pages 29-44. Morgan &
Claypool.

----

I have read the author response.

[Official Review · Reviewer 2 · rating 4 · confidence 5]
soundness 3 · originality 4 · clarity 5 · impact 4 · substance 4 · appropriateness 5 · meaningful comparison 4 · presentation format Poster

- Strengths:

The paper proposes a new method for word sense induction from synonymy
dictionaries. The method presents a conceptual improvement over existing ones
and demonstrates robust performance in empirical evaluation. The evaluation was
done thoroughly, using a number of benchmarks and strong baseline methods. 

- Weaknesses:

Just a couple of small points. I would like to see more discussion of the
nature of the evaluation. First, one observes that all models' scores are
relatively low, under 50% F1. Is there room for much improvement or is there a
natural ceiling of performance due to the nature of the task? The authors
discuss lexical sparsity of the input data but I wonder how much of the
performance gap this sparsity accounts for. 
Second, I would also like to see some discussion of the evaluation metric
chosen. It is known that word senses can be analyzed at different levels of
granularity, which can naturally affect the scores of any system.
Another point is that it is not clear how the authors obtained vectors for word
senses that they used in 3.4, if the senses are only determined after this
step, and anyway senses are not marked in the input corpora. 

- General Discussion:

I recommend the paper for presentation at the ACL Meeting. Solid work.